# CALD1 Modulates Gliomas Progression via Facilitating Tumor Angiogenesis

**DOI:** 10.3390/cancers13112705

**Published:** 2021-05-30

**Authors:** Quan Cheng, Anliu Tang, Zeyu Wang, Ning Fang, Zhuojing Zhang, Liyang Zhang, Chuntao Li, Yu Zeng

**Affiliations:** 1Department of Neurosurgery, Xiangya Hospital, Central South University, 87 Xiangya Road, Changsha 410008, China; chengquan@csu.edu.cn (Q.C.); zeyuwangn@csu.edu.cn (Z.W.); zhangliyang@csu.edu.cn (L.Z.); 2National Clinical Research Center for Geriatric Disorders, Xiangya Hospital, Central South University, 87 Xiangya Road, Changsha 410008, China; 3Department of Gastroenterology, The Third Xiangya Hospital, Central South University, 138 Tongzipo Road, Changsha 410013, China; tanganliuxy3@csu.edu.cn (A.T.); golempowers@csu.edu.cn (N.F.); 4Department of Scientific Research, Xiangya Hospital, Central South University, 87 Xiangya Road, Changsha 410008, China; zzjdew@csu.edu.cn

**Keywords:** CALD1, angiogenesis, glioma, brain tumor

## Abstract

**Simple Summary:**

Caldesmon has recently attracted attention in cancer due to its roles in cell migration, invasion and proliferation. l-CALD1 was also considered a potential serum marker for glioma. However, little is known about mechanisms underlying the effect of CALD1 on the microvascular facilitation and architecture in glioma. The purpose of this study was to explore the role of CALD1 for prediction glioma patient prognosis and in glioma angiogenesis. The findings of this study suggested that l-CALD1 could imply abnormal microvessels in anaplastic astrocytoma and GBM. In addition, high CI (calmodulin index) predicted worse prognosis in glioma, and furthermore, CALD1 may serve as a key marker for monitoring the progress of glioma and a novel target for therapy.

**Abstract:**

Angiogenesis is more prominent in anaplastic gliomas and glioblastoma (GBM) than that in pilocytic and diffuse gliomas. Caldesmon (CALD1) plays roles in cell adhesion, cytoskeletal organization, and vascularization. However, limited information is available on mechanisms underlying the effect of CALD1 on the microvascular facilitation and architecture in glioma. In this study, we explored the role of CALD1 in gliomas by integrating bulk RNA-seq analysis and single cell RNA-seq analysis. A positive correlation between CALD1 expression and the gliomas’ pathological grade was noticed, according to the samples from the TCGA and CGGA database. Moreover, higher CALD1 expression samples showed worse clinical outcomes than lower CALD1 expression samples. Biofunction prediction suggested that CALD1 may affect glioma progression through modulating tumor angiogenesis. The map of the tumor microenvironment also depicted that more stromal cells, such as endothelial cells and pericytes, infiltrated in high CALD1 expression samples. CALD1 was found to be remarkably upregulated in neoplastic cells and was involved in tumorigenic processes of gliomas in single cell sequencing analysis. Histology and immunofluorescence analysis also indicated that CALD1 associates with vessel architecture, resulting in glioma grade progression. In conclusion, the present study implies that CALD1 may serve as putative marker monitoring the progress of glioma.

## 1. Introduction

Glioblastoma (GBM) is the most lethal and malignant primary cerebral tumor [1]. In clinics, IDH mutation or wide-type, 1p19q status and O6-methylguanine-DNA methyltransferase (MGMT) promoter methylation status are used to predict patient survival and personalize therapy [2]. In pathological events, microvascular proliferation is representative. Microvascular facilitation and architecture branch or sprout proliferation in anaplastic gliomas and GBM, which is a striking contrast to pilocytic and diffuse gliomas. Tumor angiogenesis-induced perivascular edema is an important imaging feature of high-grade gliomas.

Caldesmon (CALD1) performs as cytoskeleton-associated protein and regulates cell morphology and motility via actin filaments modulation [3] The transcriptional variance of the CALD1 gene consists of 15 exons and is characterized by the recombination of alternative splicing modes: the high-molecular mass smooth muscle isoform (120–150 kd, *h-*CALD1), and the low-molecular non-muscle cells isoform (70–80 kd, *l-*CALD1) [4,5,6]. *l-*CALD1 plays roles in cell adhesion, cytoskeletal organization, and vascularization [6,7]. Different isoforms of CALD1 may character the distinct functions of cell types. Intracranial metastasis of non-small cell lung cancers (NSCLCs) presents high relevance to *l-*CALD1 upregulation, which is manipulated by nitric oxidase (NO) inducing calcium inner-fluxing [8]. It also contributes to the formation of cell adhesion structures “invadopodium” whose componential characterizations include proteases and cytoskeletons [9]. Microvascular architecture also associates to *l-*CALD1 expression levels in glioma. The density of angiogenesis in pilocytic astrocytoma demonstrates non-significance to normal; representatively, in anaplastic glioma and GBM [6]. Characterization of *l-*CALD may serve as a putative marker guiding clinical diagnosis and monitoring the progress of glioma [10].

Unfortunately, despite the wealth of information regarding CALD1 activities only in neovascularization, relatively little is known about it in tumor invasion, glioma subtypes, overall survival, and relative immunological activities. Some researchers have indicated that CALD1 expression is restricted to vessel architectures in glioma [11]. Our previous study reveals that CALD1 transcriptomic level in glioma patient-derived tumor cells (GPDCs) associating with glioma grades progression is an accompanying phenomenon [12]. In this study, we explored the role of CALD1 in gliomas by integrating bulk RNA-seq analysis and single cell RNA-seq analysis. A positive correlation between CALD1 expression and the gliomas’ pathological grade was noticed according to the samples from the TCGA and CGGA database. Moreover, higher CALD1 expression samples showed worse clinical outcomes than lower CALD1 expression samples. Biofunction prediction suggested that CALD1 may affect gliomas progression through modulating tumor angiogenesis. The map of the tumor microenvironment also depicted that more stromal cells, such as endothelial cells and pericytes, infiltrated in high CALD1 expression samples.

## 2. Materials and Methods

### 2.1. Data Collection

The mRNA sequencing data of genes encoding calmodulin-dependent proteins were downloaded from The Cancer Genome Atlas (TCGA) dataset, which was set as the training cohort. The mRNA sequencing data from the Chinese Glioma Genome Atlas (CGGA) dataset were set as the validation cohort. Corresponding clinical information was also downloaded.

### 2.2. Bioinformatic Analysis

The enrichment score of pathways form the GO/KEGG enrichment analysis was calculated by conducting the Gene Set Variation Analysis (GSVA). The Gene Set Enrichment Analysis (GSEA) analysis was performed by R packages clusterProfiler [13] and misgdf. The overall survival analysis was employed to predict the overall survival outcome of different groups. The univariate Cox regression analysis was performed to identify unfavorable biomarker. Gene correlation was calculated with Pearson’s coefficient, and exhibited with circle plot (correlation coefficient > 0.3).

Tumor immune landscape was mapped with the ESTIMATE algorithm [14], the CIBERSORT algorithm [15] and the 28-immunocytes infiltration project [16]. Stromal cell proportion was calculated with the xCell algorithm [17]. All analyses were performed by R (version 3.6.1).

### 2.3. Single Cell RNA-Sequencing

Eight single-cell RNA sequencing samples were from the Gene Expression Omnibus (GEO) under the accession number of GSE138794. scRNA-seq analysis was performed as previously described [18]. The single-cell gene expression matrix was processed using the R package Seurat. R package UMAP was applied for dimension reduction and visualization. Pseudotime trajectories reconstruction was performed using R package Monocle.

### 2.4. Cell Culture and Stable Cell Lines Selection

The U251 human glioblastoma cell line was purchased from Sigma-Aldrich. The LN229 cell line was purchased from the American Type Culture Collection (ATCC, Rockville, MD, USA). U251 and LN229 were maintained in DMEM, supplemented with 10% fetal bovine serum (FBS, Gibco, Waltham, MA, USA), 1% NEAA (100×, Gibco, Waltham, MA, USA), 1% sodium pyruvate (100×, Gibco, Waltham, MA, USA), and 1% pen strep glutamine (100×, Gibco, Waltham, MA, USA).

For CALD1 knock-out, U251 and LN229 were infected with lentiviral-based CRISPR/Cas9 knock-out vector (CRISPR KO vector from Dr. Feng Zhang lab, psPAX2 and pCMV-VSV-G from Addgene, Watertown, MA, USA). Stable cell lines were selected by puromycin at a concentration of 1 μg/mL.

### 2.5. Patients and Tissue Samples

A total of 70 adult glioma specimens were used. These specimens were obtained from patients who underwent either explorative or radical surgery at our hospital after informed consent was obtained and approved by the ethics committee of the Xiangya Hospital of Central South University (Hunan, China). Historical diagnosis of glioma was performed according to the WHO classification, including tumor specimens of WHO grade I (11 cases), WHO grade II (25 cases), WHO grade III (10 cases), and WHO grade IV (24 cases).

### 2.6. Histology and Immunofluorescence Analysis

U251/LN229 cell lines were mounted on slides for immunofluorescence analysis as previously described [8]. The primary antibody for Actin was 1:500 β-Actin (Santa Cruz, CA, USA). For the formalin-fixed paraffin-embedded (FFPE) tissue, deparaffinization, rehydration and antigen retrieval were performed as previously described [19]. After blocking with 3% BSA for 60 min, the specimens were treated overnight at 4 °C primary antibodies against the following proteins: caldesmon (rabbit, 1:400, Abcam, Cambridge, MA, USA). The samples were rinsed three times with PBS and incubated in fluorophore-conjugated secondary antibody for 1–2 h. After washing three times in PBS, the slides were mounted with ProLong Diamond Antifade Mountant with 4′,6-diamidino-2-phenylindole, cover slipped, and imaged with a fluorescence microscope (Olympus, Tokyo, Japan). Glioma tissue array slices for immunohistology (IHC) staining were performed accordingly to the manual (Maixin biotechnologies, Fuzhou, China).

### 2.7. Western Blot Analysis

Cells were solubilized in cold RIPA lysis buffer and separated with 5% SDS-PAGE. Following this, proteins were transferred from the gel to a PVDF membrane. Membranes were blocked in 5% non-fat dried milk in PBST for 2 h and then incubated overnight with L-Caldsmon1 (1:200, Santa Cruz, Dallas, TX, USA) and β-Actin (1:500, Santa Cruz, Dallas, TX, USA). After incubation with the appropriate secondary antibody, immune complexes were detected using an ECL kit. Results were visualized by autoradiography using preflashed Licor Image Studio version 5.2.

### 2.8. Transwell Assay

The upper chambers of a Transwell chamber, with a Makrolon micro-porous membrane (8-μm pore size; American Costar Company, Washington, WA, USA) were placed into 24-well plates, and 10 μg/mL type I collagen (SIGMA, St. Louis, MO, USA) was then added to these upper chambers for coating, followed by air drying. After the addition of DMEM containing 10% serum, cells maintained in serum-free DMEM were harvested and transferred into the upper chambers, followed by incubation for 12 h at 37 °C in a humidified environment of 5% CO_2_. Thereafter, the upper chambers were collected and washed in PBS. A cotton swab was used to remove the non-migrated cells, and the membrane was fixed in 4% paraformaldehyde for 10 min; 20 μL MTT was added to each plate for 4 h at 37 °C. Following the addition of DMSO, the dark blue crystals of MTT-formazan were dissolved by shaking the plates at room temperature, and the absorbance was then measured on a microplate reader (Bio-Rad, Hercules, CA, USA) using a test wavelength of 570 nm with a reference wavelength of 630 nm. Each experiment was performed three times.

### 2.9. Statistical Analysis

The log-rank test was applied to compare overall survival difference between different groups. The Student’s t-test was employed to compare two groups and ANOVA analysis was performed to compare multiple groups. *p* value < 0.05 was considered as statistical difference.

## 3. Results

### 3.1. CALD1 Promotes Gliomas Progression

CALD1 expression was increased in high grade glioma (Figure 1A,B), IDH wildtype glioma (Figure 1C), 1p19q non-codeletion glioma (Figure 1D), MGMT unmethylated glioma (Figure 1E) and aggressive subtype glioma (Figure 1F). A similar expression profile was also depicted in the CGGA dataset (Appendix A). The overall survival analysis based on the LGG GBM cohort (*p* value < 0.001; Figure 1G), the LGG cohort (*p* value < 0.001; Figure 1H) and the GBM cohort (*p* value < 0.001; Figure 1I) suggests that high CALD1 serves as promoter of glioma. Low CALD1 expression indicates worse survival outcome both in the MGMT methylated (*p* value < 0.001) and MGMT unmethylated (*p* value < 0.001; Figure 1J) glioma. High CALD1 expression implies short overall survival time in the radiotherapy accepted (*p* value < 0.001) or not-accepted (*p* value < 0.001; Figure 1K) group. Verification results from the CGGA dataset indicate that high CALD1 expression is associated with worse survival outcome (Appendix A).

Next, potential mechanisms of how CALD1 affected glioma progression were predicted by employing the GO/KEGG enrichment analysis (Figure 1L and Appendix A). High expression CALD1 was positively correlated with regulation of cell migration by vascular endothelial growth factor signaling pathway, regulation of bicellular tight junction assembly, substrate adhesion dependent cell spreading, blood vessel endothelial morphogenesis, cellular response to nitric oxide, p53 signaling pathway and JAK/STAT signaling pathway. Additionally, correlation between CALD1 expression and the results from KEGG enrichment analysis suggest that pathways such as focal adhesion, p53 signaling pathway and JAK/STAT signaling pathway may be activated in samples from the high CALD1 group (Figure 1M).

### 3.2. The Association between CALD1 and Tumor Microenvironment

The association between CALD1 and the tumor microenvironment was investigated. Stromal cells proportion in the tumor microenvironment was analyzed, along with the CALD1 expression. More fibroblasts, endothelial cells, smooth muscle and microvascular endothelial cells were observed in the microenvironment of tumor with high CALD1 expression compared with the low CALD1 expression samples (Figure 2A,B). Then, the correlation between CALD1 expression and genes associated to different GO pathways was further calculated, including regulation of endothelial cell chemotaxis, blood vessel endothelial cell proliferation involved in sprouting angiogenesis, branching involved in blood vessel morphogenesis, cellular response to nitric oxide, vascular endothelial growth factor signaling pathway and regulation of bicellular tight junction assembly (Figure 2C). As illustrated, CALD1 expression was positive correlated with the expression of VEGFA, THBS1, CDK2, CCR7, HSPB1, KDR, LGMN, NRP1, PRKD2, EPHA2, Myo1c, RUNX1 et al.

The ESTIMATE algorithm predicted more immunocytes and stromal cells ratio in high CALD1 expression samples (Appendix A). Results concluded from the CIBER-SORT algorithm suggested that macrophage and regulatory T cells were preferential when infiltrated in high CALD1 expression samples while follicular helper T cells were more common in the low CALD1 expression samples (Appendix A). However, the 28-immunocytes project implied that NK cells, macrophage, dendritic cells and T cells were positively correlated with high CALD1 expression; in the meantime, T helper cells and CD8 T cells were negatively correlated (Appendix A). Therefore, CALD1 might closely associate with the tumor microenvironment by affecting immunocyte infiltration and tumor angiogenesis. Results from the validation cohort also support the positive correlation between the expression of CALD1 and the enrichment score of immunocytes and stromal cells (Appendix A). The CIBERSORT algorithm and the 28-immunocytes project drew a similar conclusion with the training cohort (Appendix A). Therefore, CALD1 affected tumor microenvironment and was highly associated with tumor angiogenesis.

### 3.3. The Expression of CALD1 in Gliomas Was Further Investigated via Single Cell Sequencing Analysis

The expression of CALD1 in gliomas was further investigated via single cell sequencing analysis. UMAP plot depicting eight cell clusters in the GBM microenvironment, and the distribution of CALD1 expression levels within the eight cell clusters is shown in Figure 3A. CALD1 was found to be remarkably upregulated in neoplastic cells (Figure 3B). Further, the single-cell trajectory of neoplastic cells contained four main branches and identified seven cell states (Figure 3C). The pseudotime of cell states revealed that neoplastic cells travelled from branch point 2 to branch point 3, while CALD1 had a differential expression around branch point 3 (Figure 3C). The differentially expressed genes around branch point 3 were identified for performing GO enrichment analysis in biological process (BP) (Figure 3D) and molecular function (MF) (Figure 3E). The BP results confirmed that CALD1 was involved in the regulation of vascularization. The MF results confirmed that CALD1 was responsible for protein production. Moreover, KEGG enrichment analysis showed that CALD1 was involved in tumorigenic processes of gliomas (Figure 3F).

### 3.4. CALD1 Associates with Vessels Architecture Resulting Glioma Grades Progression

To investigate the function of CALD1 in GBM cells, we confirm the expression and location of l-CALD1 in GBM cell lines. We found that l-CALD1 was located in cell membrane and presented highlighting and dense fluorescence in cell polarization. Furthermore, l-CALD1 was co-localized with F-actin in GBM cell lines (Figure 4A,B). According to the previous data, dysfunctional l-CALD1 may lead to a decline in cell mobility. We further performed the CRIPR/Cas9, targeting l-CALD1 in LN229 and U251 cell lines. In protein level, the result presented efficiently l-CALD1 knock-out (Figure 4F). Dramatically, knocked-out l-CALD1 inhibited LN229 and U251 mobilities (Figure 4C,G). Morphologic observations and clinicopathologic correlations demonstrated that the l-CALD1 expression was restricted to blood vessels. In pilocytic and diffusion astrocytoma, l-CALD1 were stained and located in microvessels with the normal vascular morphology. In contrast, microvessel abnormalities were detected and observed in anaplastic astrocytoma and GBM (Figure 4D,E,H,I). Considered together, these results indicate that l-CALD1 potentially presents the validation of glioma grading determination.

### 3.5. Expression Profile of Genes Encoding Calmodulin-Dependent Proteins

In order to analyze the role of CALD1 in glioma comprehensively, we selected genes which were calmodulin-dependent proteins differentially expressed in gliomas with different features. In the TCGA dataset, CALD1, CALML4, CALML6, CALM1 and CALM3 mostly enriched in GBM. In the meantime, CALML5, CALM3 and CALML3 were expressed in LGG. Notably, the expression of CALML5 and CALML3 was barely detected in GBM by comparing to LGG (Appendix A). As for the validation cohort, CALD1, CALML4 and CALML6 were expressed in GBM. However, no significantly difference of CALM3 expression was observed between LGG and GBM. CALM1 and CALM2 were expressed in LGG instead of GBM, as with the training cohort. Besides, the expression of CALML5 and CALML3 was not found in the validation cohort (Appendix A).

Different expression profiles were observed between WHO grade II and WHO grade III glioma. Results from both the TCGA and CGGA dataset suggest that the expression of CALD1, CALML4 and CALML6 was higher in WHO grade III glioma than WHO grade II glioma. Besides, no difference of CALM1 and CALM2 expression was noticed (Appendix A).

Considering the status of IDH can assist in predicting patients’ overall survival outcome, its association with calmodulin-dependent proteins was also explored. High expressions of CALD1, CALML4, CALML6, CALM1 and CALM2 were associated with IDH wildtype glioma. The expressions of CALML5, CALM3 and CALML3 were upregulated in IDH mutant glioma (Appendix A). However, in the CGGA dataset, only CALD1, CALML4 and CALML6 were associated with IDH wildtype glioma, and no specific relationship with CALM3, CALM1 and CALM2 expression was noticed (Appendix A). In general, the expression of genes encoding calmodulin-dependent proteins was connected to glioma pathological grades and IDH status, implying their role in glioma progression. CALD1, CALML4 and CALML6 might serve as unfavorable biomarkers in glioma. Therefore, in order to integrally analyze their role in glioma progression, the calmodulin index (CI) model was constructed.

### 3.6. The Calmodulin Index Model

The CI model based on the expression of genes encoding calmodulin-dependent proteins was built by performing the GSVA analysis. Univariate Cox regression analysis demonstrated that CALD1 has the highest hazard ratio, compared with other members, which implied that CALD1 played a critical role in the CI model (Table 1). The distribution of CI is illustrated with heatmap, along with the expression of genes encoding calmodulin-dependent proteins and corresponding clinical features (Figure 5A). As illustrated, samples with high expression of genes encoding calmodulin-dependent proteins were associated with high CI. A similar distribution was also noticed in the CGGA dataset (Appendix A). In the meantime, high CI was related to high grade glioma (Figure 5B,C), IDH wildtype glioma (Figure 5D), 1p19q non-codeletion glioma (Figure 5E), MGMT unmethylated glioma (Figure 5F) and aggressive subtype glioma (Figure 5G). Results from the validation cohort also support that observation (Appendix A).

The overall survival analysis suggested that high CI samples manifested worse survival outcomes relative to low CI in the LGGGBM cohort (*p* value < 0.0001; Figure 5H) and LGG cohort (*p* value = 0.0043; Figure 5I), but no difference in the GBM cohort (*p* value = 0.13; Figure 5J). In the CGGA dataset, the overall survival analysis on the LGGGBM cohort (*p* value = 0.0015; Appendix A), LGG cohort (*p* value = 0.41; Appendix A) and the GBM cohort (*p* value = 0.31; Appendix A) indicated the CI model can predict glioma patient survival outcome.

The MGMT promoter methylation status can predict response to temozolomide, a common chemotherapeutic drug for glioma, in patients with glioma. In MGMT methylated glioma from the TCGA dataset, high CI samples manifested better survival outcome than low CI samples (*p* value = 0.0013; Figure 5K). MGMT unmethylated glioma with higher CI also showed better prognosis than with lower CI samples (*p* value < 0.0001; Figure 5K). These results obviously conflict with the previous discovery that high CI was recognized as an unfavorable signature for glioma. This result indicated that calmodulin-dependent proteins might be associated with tumor sensitivity to temozolomide. For radiotherapy, the median overall survival time of high CI samples was significantly shorter than low CI samples, either in the radiotherapy accepted group (*p* value < 0.001) or the radiotherapy not-accepted group (*p* value < 0.001; Figure 5L). Verification on the CGGA dataset also showed similar results (Appendix A). Thus, the CI model was able to predict glioma patient survival. The expression of genes encoding calmodulin-dependent proteins was almost upregulated in high CI samples, whereas not all genes’ expression increased in aggressive glioma, such as GBM or IDH wildtype glioma, indicating their probable different contribution to the CI model.

### 3.7. The GO/KEGG Enrichment Analysis and the GSEA Analysis Based on the CI Model

The GO/KEGG enrichment analysis and the GSEA analysis were performed to analyze the difference between high and low CI samples. High CI samples were associated with cell migration involved in sprouting angiogenesis and regulation of endothelial cell chemotaxis, VEGF signaling pathway, according to the GO/KEGG enrichment analysis (Figure 6A,B). The GSEA analysis suggested that cell migration involved in sprouting angiogenesis, the regulation of vascular smooth muscle cell proliferation and the negative regulation of vascular endothelial cell proliferation were all connected to high CI samples (Figure 6C). In summary, the CI model can be applied to predict patients’ prognosis, and high CI samples showing worse survival outcomes might result from the activation of tumor angiogenesis.

## 4. Discussion

Caldesmon has recently attracted attention in cancer due to its roles in cell movement, such as migration, invasion and proliferation [20,21]. *l-*CALD1 presents in many cell types, except muscle and performance, an essential role in regulating actin dynamics. *l-*CALD1 can be a biomarker for the pathological diagnosis of cancers because the upregulated expression of *l-*CALD1 has been observed in different cancer types [22,23,24]. *l-*CALD1 is also considered as a potential serum marker for glioma [10]. *l-*CALD1 can be a biomarker for prediction of the chemoradiotherapy response. Kim et al. found that *l-*CALD1 can decrease the chemoradiotherapy susceptibility of human colon cancer cells [22].

In this work, we investigated the association between *l-*CALD1 and glioma. Increased *l-*CALD1 expression promoted glioma progression by promoting tumor angiogenesis and immunocytes infiltration. More stromal cells such as endothelial cells infiltrated in high *l-*CALD1 expression samples further supported the connection between *l-*CALD1 and tumor angiogenesis. Further, the expression level of *l-*CALD1 was explored in the tumor microenvironment of glioma via single-cell sequencing analysis. Notably, *l-*CALD1 was found to be more elevated in neoplastic cells and vascular cells. Based on the differentially expressed genes around branch point 3, GO enrichment results in BP and MF confirmed that *l-*CALD1 played a critical role in the angiogenetic process and tumorigenic process of glioma. KEGG enrichment results also proved that *l-*CALD1 was involved in several tumorigenic signaling pathways. These results suggest that increased *l-*CALD1 expression promoted tumor progression through modulating tumor angiogenesis. Though other research has found that *h-*CALD1 appears to be the most specific and sensitive marker for vessel wall detection [25], we believe that *l-*CALD1 can imply abnormal microvessels in anaplastic astrocytoma and GBM.

Innovatively, we also proposed a scoring system, the CI model, based on the GSVA analysis according to genes encoding calmodulin-dependent proteins. As illustrated, high CI samples carry with malignancy clinical features such as high WHO grade, wildtype IDH, unmethylated MGMT, non-codeletion 1p19q. Besides, proneural glioma showed the best survival outcome compared to the other two subtypes, including mesenchymal and classical. The overall survival analysis demonstrates that high CI is associated with shorter survival time. However, the expression of CALML5 was missed in the CGGA cohort. Considering its expression in GBM from the training cohort was also missed, the involvement of this gene in the CI model may not affect its accuracy. As expected, the validation of this model further supported the high accuracy of this model. Therefore, high CI is recognized as an unfavorable feature of glioma.

Single-cell RNA sequencing (scRNA-seq) comprehensively describes cell types and cellular states within both normal and disease tissues [26], and also the landscape of immune phenotypes in tumor microenvironment. Unfortunately, the insufficiently acknowledged benefits of the *l-*CALD1 expression and distribution exist for GBM and tumor microenvironment. Innovatively, this study employed scRNA-seq analysis to reveal the expression of l-CALD1 in gliomas. We found a UMAP plot depicting eight cell clusters in GBM microenvironment, and the distribution of *l-*CALD1 expression level within the eight cell clusters. Notably, *l-*CALD1 was found to be remarkably upregulated in neoplastic cells. Further, single-cell trajectory of neoplastic cells contained four main branches and identified seven cell states. The pseudotime of cell states revealed that neoplastic cells traveled from branch point 2 to branch point 3, and that *l-*CALD1 had a differential expression around branch point 3. The BP results confirmed that *l-*CALD1 was involved in the regulation of vascularization. The MF results confirmed that *l-*CALD1 was responsible for protein production.

## 5. Conclusions

The current study suggests that l-CALD1 could imply abnormal microvessels in anaplastic astrocytoma and GBM, it recognizes high CI as an unfavorable feature of glioma, and find that CALD1 could serve as putative marker monitoring the progress of glioma.

## Figures and Tables

**Figure 1 cancers-13-02705-f001:**
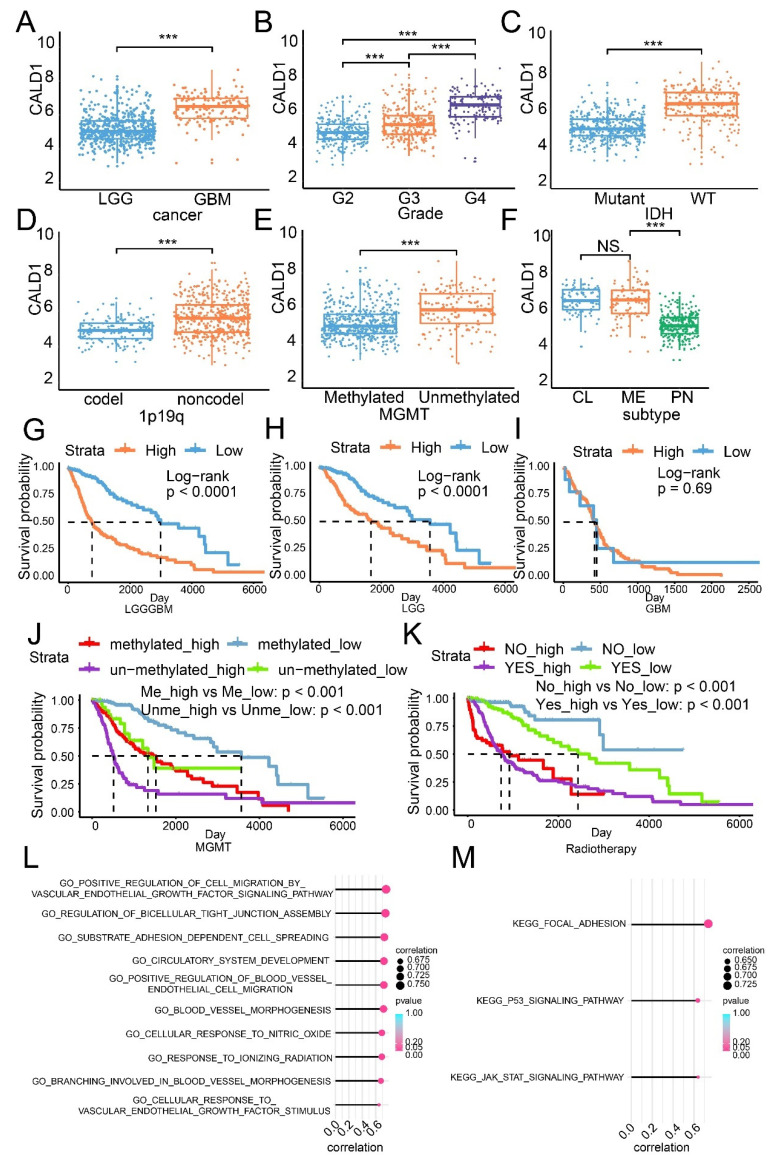
The expression profile, the overall survival analysis and pathways prediction based on CALD1 expression in the TCGA dataset. The distribution of CALD1 in cancer (**A**), WHO grade (**B**), IDH status (**C**), 1p19q status (**D**), MGMT status (**E**) and glioma subtype (**F**). The overall survival analysis based on the LGGGBM cohort (**G**), the LGG cohort (**H**), the GBM cohort (**I**), MGMT status (**J**) and radiotherapy (**K**). (**L**,**M**) The correlation between CALD1 expression and pathways from the GO/KEGG enrichment analysis. NS: no statistical difference; ***: *p* value < 0.001.

**Figure 2 cancers-13-02705-f002:**
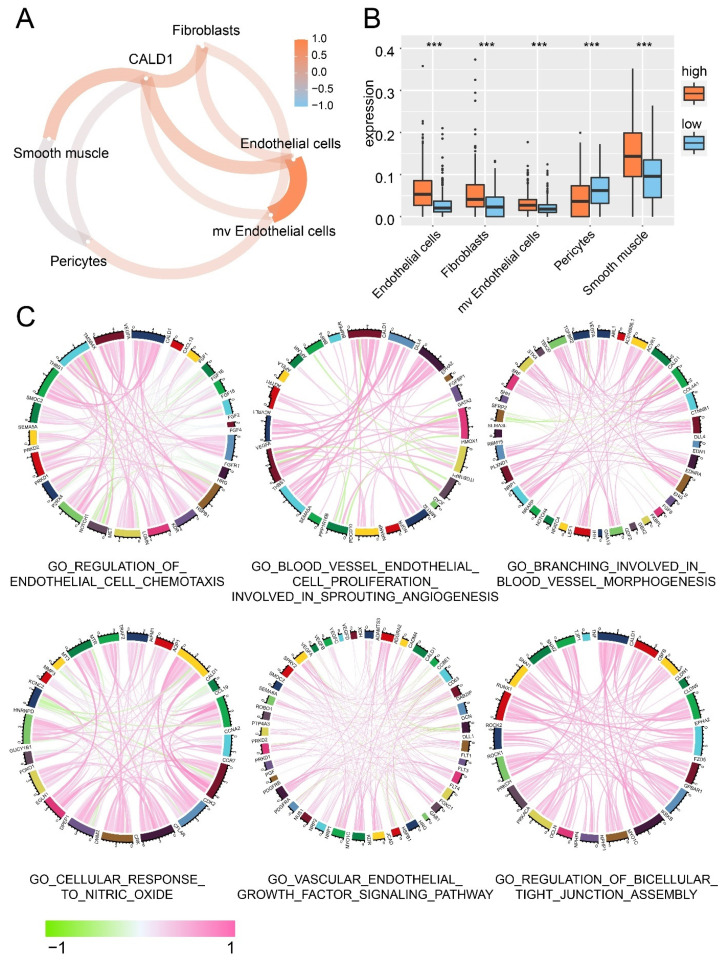
Stromal cells proportion in tumor microenvironment. (**A**) The correlation between CALD1 expression and the enrichment score of stromal cells. (**B**) The distribution of stromal cells in high and low CALD1 expression samples. (**C**) The correlation between the expression of CALD1 and genes from GO pathways. NS: no statistical difference; ***: *p* value < 0.001.

**Figure 3 cancers-13-02705-f003:**
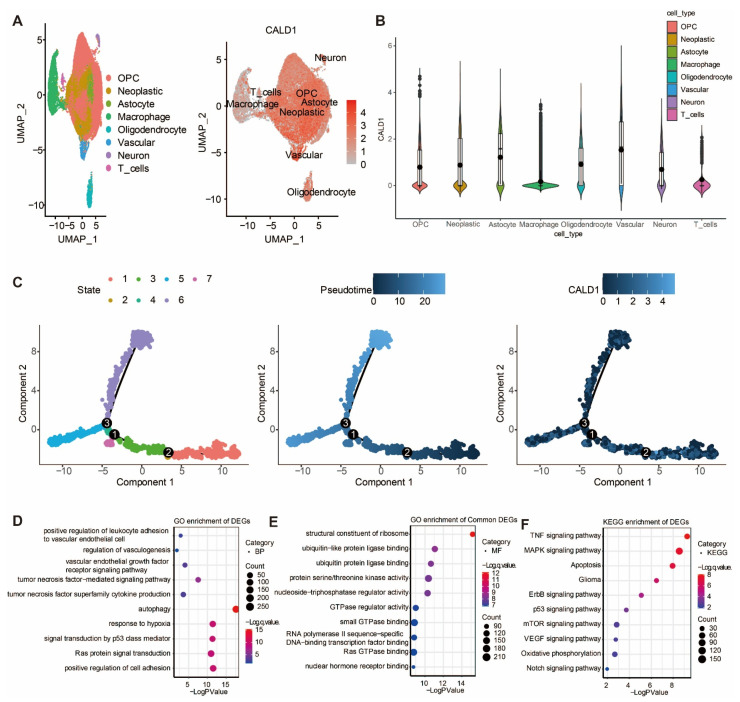
Single cell sequencing analysis in GBM microenvironment. (**A**) UMAP plot depicting eight cell clusters and the distribution of CALD1 expression levels within the eight cell clusters. (**B**) Violin plot exhibiting the relative expression level of CALD1 in eight cell clusters. (**C**) The single-cell trajectory of neoplastic cells contains four main branches. Cells are colored based on state (left), pseudotime (middle), and CALD1 (right). (**D**) GO enrichment analysis of CALD1 in BP in neoplastic cells. (**E**). GO enrichment analysis of CALD1 in MF in neoplastic cells. (**F**). KEGG enrichment analysis of CALD1 in neoplastic cells.

**Figure 4 cancers-13-02705-f004:**
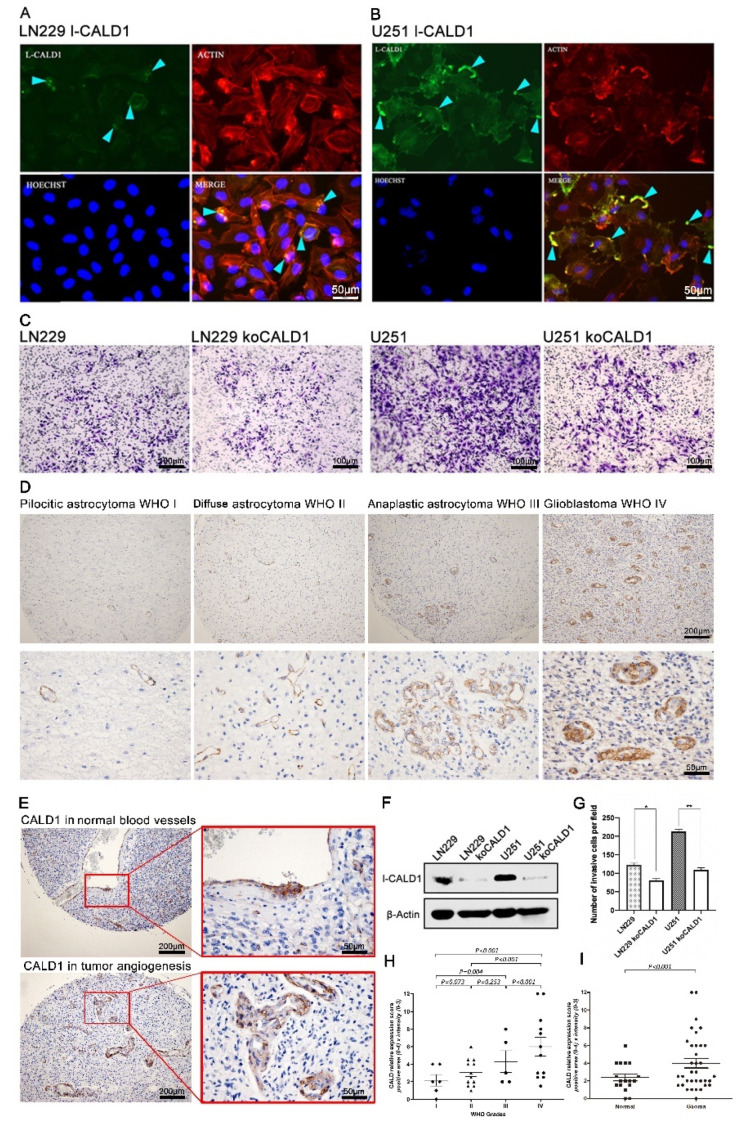
CALD1 facilitates GBM cells moveability in vitro and associates with tumor progression in pathological angiogenesis. (**A**) Immunofluorescent staining of l-CALD1 in cell line LN229. (**B**) Immunofluorescent staining of l-CALD1 in cell line U251. (**C**) Knock-out l-CALD1 affected mobility of GBM cells. (**D**) IHC staining of l-CALD1 in glioma cases according to WHO grades progression. (**E**) Comparing l-CALD1 in normal blood vessels and tumor angiogenesis. (**F**) Measurement of CRISPR/Cas9 knock-out l-CALD1 efficiency in U251 and LN229 by protein level. (**G**) Statistics of GBM cells mobility after koCALD1. (**H**) Statistics of comparing l-CALD1 expression level according to WHO grade progression. (**I**) Statistics of comparing l-CALD1 expression level in normal blood vessels and tumor angiogenesis. *: *p* value < 0.05; **: *p* value < 0.01.

**Figure 5 cancers-13-02705-f005:**
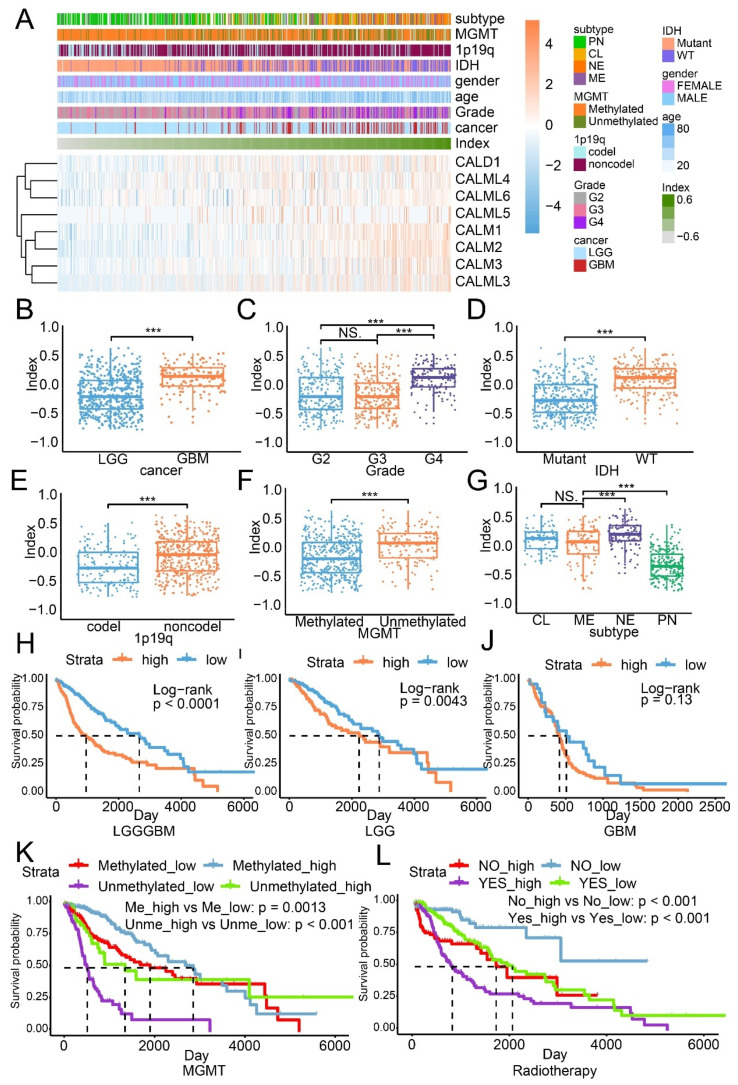
The CI model based on genes encoding calmodulin-dependent proteins in the TCGA dataset. (**A**) The distribution of CI was illustrated by a heatmap, along with clinical features and the expression of genes encoding calmodulin-dependent proteins. The association with cancer (**B**), WHO grade (**C**), IDH status (**D**), 1p19q status (**E**), MGMT status (**F**) and glioma subtype (**G**). The overall survival analysis based on the LGGGBM cohort (**H**), the LGG cohort (**I**) and the GBM cohort (**J**). The overall survival analysis based on the subgroup of MGMT status (**K**) and radiotherapy (**L**). NS: no statistical difference; ***: *p* value < 0.001.

**Figure 6 cancers-13-02705-f006:**
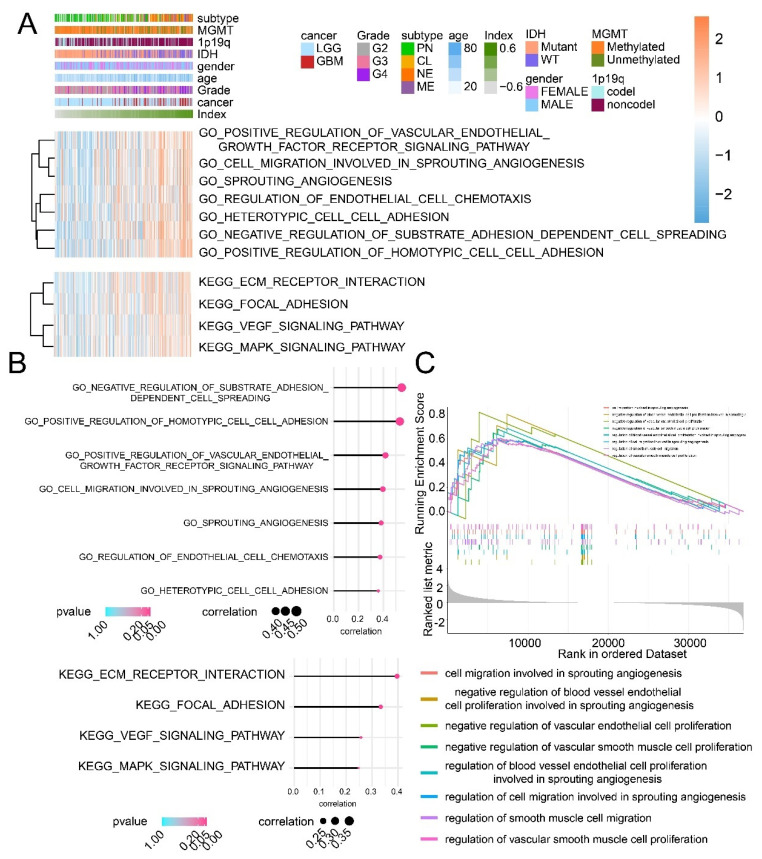
The GO/KEGG enrichment analysis and the GSEA analysis based on the CI model. (**A**) The relationship between CI and enrichment score of pathways from the GO/KEGG enrichment analysis. (**B**) The correlation between CI and pathways from the GO/KEGG enrichment analysis. (**C**) The GSEA analysis based on high and low CI samples.

**Table 1 cancers-13-02705-t001:** Univariate Cox analysis based on genes of genes encoding calmodulin-dependent proteins.

Genes	Hazard Ration	*p* Value	HR (95%CI)
CALM1	1.375	0.001817	1.375 (1.126–1.679)
CALM2	2.013	1.21 × 10^−7^	2.013 (1.553–2.608)
CALM3	0.599	1.44 × 10^−6^	0.599 (0.487–0.738)
CALML3	0.867	1.63 × 10^−7^	0.867 (0.822–0.915)
CALML4	1.888	1.70 × 10^−14^	1.888 (1.605–2.221)
CALML5	0.903	0.005257	0.903 (0.841–0.97)
CALML6	1.257	6.38 × 10^−9^	1.257 (1.163–1.357)
CALD1	2.401	4.87 × 10^−37^	2.401 (2.098–2.748)

## Data Availability

The datasets generated and analyzed during the current study are available from the corresponding author on reasonable request.

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
