# Peer review of "CALD1 Modulates Gliomas Progression via Facilitating Tumor Angiogenesis"

_cancers, 2021, doi:10.3390/cancers13112705_

Round 1

Reviewer 1 Report

The current work is nicely executed in terms of the data presented. 

Overall it is technically and scientifically sound providing convincing evidence for the proof of concept.

Minor concern:

Provide scale bars in Figure 4.

Author Response

Minor concern:

  • Provide scale bars in Figure 4.

We appreciate reviewer’s comment. We’ve added scale bars in Figure4 accordingly.

Reviewer 2 Report

Cheng Q et al analyzed two different databases for the expression and potential function of Caldesmon (CALD1). They found overexpression of CALD1 in malignant gliomas using different classifications and verified expression within cell lines. The molecule seems interesting concerning glioma progression. Described data are clearly presented in the Figures.

Major concerns

  • I think the statement in the title about the relation of CALD1 and tumor angiogenesis/progression is not really proven. The study showed database analyses and developed further models to identify connection between CALD1 and tumor progression. However, no in vivo or in vitro models were used to verify observed findings by the databases. Respective experiments are needed to validate the demonstrated coherences.
  • Please revise grammar and spelling (including: diffusion gliomas (diffuse gliomas?), vascularis (vascularization?). Sometimes reading and understanding is difficult due to grammar mistakes.
  • Materials and Methods are inadequate. A lot of information is lacking:
  1. 4B/C: Is this a Western Blot? Description/used antibodies are completely missing.
  2. 4D: Mobility assay? Boyden chamber? Description is missing.
  3. 4E-H: Did you use patient material? Ethical approval? Which patients, how many patients, patients’ characteristics? Processing of patient samples (paraffin?) Staining protocol for sections?
  4. Staining of cell lines in vitro: Actin antibody is missing
  5. Single Cell RNA-Sequencing: any information is missing. Which samples did you use for analyses? Processing of tissue/cells and RNA.
  6. TCGA-Database: Which data set was used for analyses?
  • 4: How do you define “normal” blood vessels?, How do you identify blood vessels? You can use CD31 for vessel detection.
  • It is difficult to determine a specific CALD1 expression on vessels if no marker for vasculature/endothelial cells is stained.

Minor concerns:

  • Term “multiforme” is no longer used
  • Abbreviations must be explained, e.g. NSCLC, MF, BP, GSVA, CI, non-codel (non-codeletion?), “mv” (microvascular?) endothelial cells
  • Figure 1M is not referenced in the text
  • Table 1 was not included in the manuscript

Author Response

Major concerns

  • I think the statement in the title about the relation of CALD1 and tumor angiogenesis/progression is not really proven. The study showed database analyses and developed further models to identify connection between CALD1 and tumor progression. However, no in vivo or in vitro models were used to verify observed findings by the databases. Respective experiments are needed to validate the demonstrated coherences.

We appreciate the reviewer’s thoughtful comments. In this study, we have provided the functional investigation of CALD1 in GBM cell lines in vitro. Accordingly, the results demonstrated that L-CALD1 aggerated in polarized GBM cells. Furthermore, the based expression levels of CALD1 in GBM cell lines U251 and LN229 were significant difference that indicated relevance to cell mobilities. Knocking-out CALD1 in GBM cell lines strongly attenuated cellular migration. However, we have noted that these in vitro respective experiments also had weaknesses. Therefore, bio-informatic analysis which based on the giant cohorts’ datasets and creative analysis perspectives encouraged the investigation of CALD1 functions in GBM and angiogenic progress. Additionally, we have addressed reviewers’ comments to make the conclusion and figures clear. Thanks for the comments again.

  • Please revise grammar and spelling (including: diffusion gliomas (diffuse gliomas?), vascularis (vascularization?). Sometimes reading and understanding is difficult due to grammar mistakes.

We appreciate such detailed review comment. We have changed diffusion gliomas to diffuse gliomas, vascularis to vascularization.

  • Materials and Methods are inadequate. A lot of information is lacking:
  1. 4B/C: Is this a Western Blot? Description/used antibodies are completely missing.

We have added Western blot analysis in Materials and Methods. Performed antibodies in this study were addressed in revised version of manuscript: “Western blot analysis. Cells were solubilized in cold RIPA lysis buffer and separated with 5% SDS-PAGE. Following this, proteins were transferred from the gel to a PVDF membrane. Membranes were blocked in 5% non-fat dried milk in PBST for 2h and then incubated overnight with L-Caldsmon1 (1:200, Santa Cruz, USA), β-Actin (1:500, Santa Cruz, USA) After incubation with the appropriate secondary antibody, immune complexes were detected using an ECL kit. Results were visualized by autoradiography using preflashed. Licor Image Studio version 5.2.” 

  1. 4D: Mobility assay? Boyden chamber? Description is missing.

Boyden chamber was used. We have added Transwell assay in Materials and Methods. “Transwell assay The upper chambers of a Transwell chamber, with a Makrolon micro-porous membrane (8-μm pore-size; American Costar Company) were placed into 24-well plates, and 10 μg/mL type I collagen (SIGMA, St. Louis, MO, USA) was then added to these upper chambers for coating, followed by air drying. After the addition of DMEM containing 10% serum, cells maintained in serum-free DMEM were harvested and transferred into the upper chambers, followed by incubation for 12h at 37°C in a humidified environment of 5% CO2. Thereafter, the upper chambers were collected and washed in PBS. A cotton swab was used to remove the non-migrated cells, and the membrane was fixed in 4% paraformaldehyde for 10 min; 20 μL MTT was added to each plate for 4h at 37°C. Following the addition of DMSO, the dark blue crystals of MTT-formazan were dissolved by shaking the plates at room temperature, and the absorbance was then measured on a microplate reader (Bio-Rad, Hercules, CA, USA) using a test wavelength of 570 nm with a reference wavelength of 630 nm. Each experiment was performed three times.”

  1. 4E-H: Did you use patient material? Ethical approval? Which patients, how many patients, patients’ characteristics? Processing of patient samples (paraffin?) Staining protocol for sections?

We used patient material and got approval from the ethics committee of the Xiangya Hospital of Central South University. The patients and tissue information have been added in Materials and Methods.

Patient samples were paraffin embedded, the staining protocol was added in the Histology and immunofluorescence analysis section.

  1. Staining of cell lines in vitro: Actin antibody is missing

The Actin antibody for immunofluorescence was used β-Actin (1:500, Santa Cruz, USA) we have added the details in the Histology and immunofluorescence analysis section.

  1. Single Cell RNA-Sequencing: any information is missing. Which samples did you use for analyses? Processing of tissue/cells and RNA.

Thank you for your comment. The missed information of sc-RNA seq analysis has been added to the method section. 8 single-cell RNA sequencing samples were from the Gene Expression Omnibus (GEO) under the accession number of GSE138794.scRNA-seq analysis was performed as previously described. The single-cell gene expression matrix was processed using the R package Seurat. R package UMAP was applied for dimension reduction and visualization. Pseudotime trajectories reconstruction was performed using R. package Monocle.

  1. TCGA-Database: Which data set was used for analyses?

We appreciate reviewer’s comment. The mRNA sequencing data of genes encoding calmodulin dependent proteins was downloaded from The Cancer Genome Atlas (TCGA) dataset.

  • 4: How do you define “normal” blood vessels? How do you identify blood vessels? You can use CD31 for vessel detection.

Thank you for your great suggestion. Tumor blood vessels are different from “normal” counterparts in morphology, microvascular density, architecture, they are assumed to be structurally and functionally abnormal.

CD31 is a very good indicator for vessel detection. In the current study, we defined blood vessels by morphology and structure under microscope.

It is difficult to determine a specific CALD1 expression on vessels if no marker for vasculature/endothelial cells is stained.

We appreciate the critical comment by the reviewer. We believe that the comment is very pervasive to add a vasculature/endothelial cell marker. However, it is recognized that the vasculature (normal or tumor angiogenesis) identifying can be determined by obvious cavity characteristics and pathological morphology. Additionally, CD31 can be performed as endothelial marker. However, tumor angiogenesis is more complicate formalization than normal vascular. In this study, we aimed the expression and function of CALD1 in GBM tumor cells. Please consider that the staining of vessels markers was not the primary consideration for this study.

Minor concerns:

Term “multiforme” is no longer used

We appreciate reviewer’s comment. We have deleted “multiforme”.

  • Abbreviations must be explained, e.g. NSCLC, MF, BP, GSVA, CI, non-codel (non-codeletion?), “mv” (microvascular?) endothelial cells

We appreciate reviewer’s comment. We added the explanation of the abbreviations the first time it was used in the manuscript. Non-small cell lung cancer (NSCLC), molecular function (MF), biological process (BP), Gene set variation analysis (GSVA), calmodulin index (CI), non-codeletion (non-codel) and mv changed to microvascular.

We organized all the abbreviations in the section after the figure legends.

  • Figure 1M is not referenced in the text

We appreciate reviewer’s comment. We have addressed the suggestion and description of Figure 1M in manuscript as following: Correlation between CALD1 expression and the results from KEGG enrichment analysis suggested that pathways like focal adhesion, p53 signaling pathway and JAK/STAT signaling pathway may be activated in samples from high CALD1 group.

  • Table 1 was not included in the manuscript

We appreciate reviewer’s comment. Table 1 was demonstrated in Results (The calmodulin index model) section, this table is to show the list of genes encoding calmodulin dependent proteins.

Reviewer 3 Report

In this publication, Cheng and co-authors describe the role of CALD1 in gliomas progression and in tumor angiogenesis, using public databases and experimental procedures.

This is an interesting study, with many bioinformatics analysis, and some experiments based on GBM cell lines. However, this study is mainly descriptive and lacks of strong in vitro results.

This article could be considered for publication in Cancers after responding to the points below.

Major points :

  • Figure 1: it is now admitted that neural subtype is not considered as glioblastoma, please reshape Fig1F accordingly.
  • Figure 4: Images are blurry in A, please replace. Western-blots in B and C are not corresponding, bands of I-CALD1 are lower in B than in C. Add images of KO CALD1 cells with anti-CALD1 antibodies. Add quantifications of migration experiments in D.
  • How do you explain that I-CALD1 is expressed in GBM cell lines but not in tumors (Figure 4E-F).

Minor points:

  • design of Figures is not adapted is many cases : e.g. Figure 3 should be reshaped. No need of including all the dots in Fig3B.
  • Text needs to be edited carefully (e.g. CRIPR-Cas9, dots are not correctly located in sentences...)

Author Response

Major points:

  • Figure 1: it is now admitted that neural subtype is not considered as glioblastoma, please reshape Fig1F accordingly.

We appreciate reviewer’s comment. We have deleted neural subtype and reshaped Fig1F

  • Figure 4: Images are blurry in A, please replace. Western-blots in B and C are not corresponding, bands of I-CALD1 are lower in B than in C. Add images of KO CALD1 cells with anti-CALD1 antibodies. Add quantifications of migration experiments in D.

Thank you for your valuable suggestions. We apologize for the confusion. Our purpose is not to compare the expression level of l-CALD1 in parental GBM cell lines. Figure 4B and 4C were generated from different time points of ECL exposure. Obviously, l-CALD1 is much higher in U251 than LN229. For over exposure exemption, we have engaged a precise time point for result derived. We performed LN229 vs LN229 koCALD1, and U251 vs U251 koCALD1 as well.

To consider the valuable suggestion, we have re-edited the Figure 4.

  • How do you explain that I-CALD1 is expressed in GBM cell lines but not in tumors (Figure 4E-F)?

That’s a very good question. We did the tumor tissue IHC staining of I-CALD1, and also found tumor cells in high-grade tissues were positive. (supplement figure 7), selected pictures were demonstrated to emphasize the positive staining of I-CALD1 in tumor blood vessels.

Minor points:

  • design of Figures is not adapted is many cases: e.g. Figure 3 should be reshaped. No need of including all the dots in Fig3B.

We appreciate reviewer’s comment. We have reshaped Figure3 and changed the graph of 3B as following:

New version of Figure 3B.

  • Text needs to be edited carefully (e.g. CRIPR-Cas9, dots are not correctly located in sentences...)

Thank you for your kindly recommendation. We have addressed your suggestion and changed CRIPR-Cas9 to CRIPR/Cas9 in manuscript.

Round 2

Reviewer 2 Report

Based on addition of more materials and methods, the manuscript is now more understandable.

All concerns were addressed by the authors.

Reviewer 3 Report

Thank you for taking into account all my comments.

I consider now this manuscript ready for publication.